# Uncertainty-Guided Checkpoint Selection for Reinforcement Finetuning of Large Language Models

## Abstract

Reinforcement learning (RL) finetuning is crucial to aligning large language models (LLMs), but the process is notoriously unstable and exhibits high variance across model checkpoints. In practice, selecting the best checkpoint is challenging: evaluating checkpoints on the validation set during training is computationally expensive and requires a good validation set, while relying on the final checkpoint provides no guarantee of good performance. We introduce an uncertainty-guided approach for checkpoint selection (UGCS) that avoids these pitfalls. Our method identifies hard question–answer pairs using per-sample uncertainty and ranks checkpoints by how well they handle these challenging cases. By averaging the rewards of the top-uncertain samples over a short training window, our method produces a stable and discriminative signal without additional forward passes or significant computation overhead. Experiments across three datasets and three LLMs demonstrate that it consistently identifies checkpoints with stronger generalization, outperforming traditional strategies such as relying on training or validation performance. These results highlight that models solving their hardest tasks with low uncertainty are the most reliable overall.

## 1 Introduction

Large language models (LLMs) have demonstrated remarkable capabilities in solving complex reasoning tasks, particularly when fine-tuned using reinforcement learning from human feedback (RLHF) (Ouyang et al., 2022; Lee et al., 2023) and verifiable rewards (Guo et al., 2025; Liu et al., 2025b). However, the RL fine-tuning process is notoriously unstable, often leading to fluctuating performance during training. As a result, practitioners typically save multiple checkpoints throughout training, yet their quality can vary drastically. Some checkpoints exhibit strong generalization, while others, even from nearby steps, perform poorly, reflecting the high variance inherent in RL finetuning (Le et al., 2025). Selecting the optimal checkpoint, typically the one that generalizes best to unseen tasks, is a critical yet challenging step. Current practice normally depends on either training/validation performance or brute-force search over all saved checkpoints. Each of these strategies has clear limitations. Using training reward statistics is cheap, yet cannot capture true generalization, leading to "reward hacking" or overfitting checkpoints that look strong during training but perform poorly downstream (Stiennon et al., 2020; Chen et al., 2025). Validation-based model selection requires additional inference costs and suitable held-out data (Liu et al., 2025a), which may be costly or unavailable in practice. Exhaustively evaluating all checkpoints can find a strong model, but it is computationally expensive and assumes access to downstream test tasks. These challenges raise an important research question: *How can we identify training checkpoints with strong generalization capability without relying on prohibitive evaluation costs?*

To address this, we propose a new, simple, and validation-free method for LLM checkpoint selection that leverages per-sample uncertainty and reward signals already available in training logs. Our method, dubbed *Uncertainty-Guided Checkpoint Selection (UGCS)*, defines a metric to measure the quality of any LLM checkpoint during training. Prior work has shown that uncertainty is a valuable signal for improving LLM reliability, with applications in hallucination detection, enhancing factuality, and truthfulness (Manakul et al., 2023; Farquhar et al., 2024; Chang et al., 2024). These works suggest that uncertainty captures model behaviors and capabilities more faithfully than raw

accuracy, making it a stronger indicator of generalization beyond the training distribution. Building on this insight, we pioneer using uncertainty as a criterion for LLM checkpoint selection in RL finetuning, aiming to better capture generalization under unstable, high-variance training.

In particular, we first define a training window of the last $\delta$ steps. Within this window, we identify a set of the "hardest" samples—those that the model currently finds most uncertain—and evaluate checkpoint quality based on the training rewards over this set. Training samples in the window are first ranked by their uncertainty under the current model, measured by log-probabilities of the LLM outputs. Then, the top-$p\%$ most uncertain (i.e., hardest) samples within a $\delta$-step window are selected. The logged training rewards on these hardest samples are then averaged to produce a quality score for the checkpoint. This design is motivated by the observation that sustained improvements on challenging cases are a stronger indicator of model robustness than performance on easy examples (Jain et al., 2024). In contrast to global averages of training rewards, our adaptive procedure highlights per-sample variability and focuses on cases where models tend to struggle most. For instance, in mathematical tasks, some queries require multi-step reasoning, which are often underrepresented in average reward metrics. By emphasizing these difficult examples, the method can better detect whether a checkpoint has truly learned to generalize beyond routine or high-frequency cases. This approach also mitigates the risk of selecting checkpoints that appear strong overall but fail on rare or complex cases, providing a more discriminative and stable signal for checkpoint quality. Crucially, our method reuses signals already logged during training, adding negligible computational overhead.

We instantiate this approach by fine-tuning several LLMs on diverse math reasoning datasets and evaluating them on multiple benchmarks spanning different difficulty levels. Our results show that checkpoint selection via aggregating the rewards of top uncertain samples consistently matches or outperforms validation-based methods, while being simpler and more efficient. Our contributions are threefold: (i) We introduce a validation-free checkpoint selection criterion that adaptively focuses on the hardest samples; (ii) We demonstrate its effectiveness across three training datasets and four reasoning benchmarks, with gains up to 7.5% on challenging evaluations such as AMC 2023, all at negligible additional cost; (iii) Our ablations show that hardest-sample filtering and short $\delta$-step windows drive most of the gains, offering a practical, low-cost alternative to heuristics or reward-only filtering for efficient checkpointing and adaptive stopping.

## 2 RELATED WORK

Our work intersects with several areas in machine learning, including reinforcement learning for LLMs, checkpoint selection strategies, and sample difficulty estimation.

**Reinforcement Learning for LLMs**. RLHF has become a cornerstone for aligning LLMs with human preferences, as pioneered in works like InstructGPT (Ouyang et al., 2022). More recent work has explored RL with verifiable or structured rewards, where reward signals can be independently validated or computed via automated evaluation metrics (Mu et al., 2024; Guo et al., 2025; Liu et al., 2025b). These methods aim to improve LLM reasoning via RL finetuning, but they still face high variance during training, especially for tiny LLMs (Le et al., 2025), making effective checkpoint selection critical. Our method addresses this by providing a fine-grained selection criterion directly from training logs, complementing recent efforts in stabilizing RL finetuning, such as reward shaping (Chen et al., 2025) and iterative fine-tuning (Yuan et al., 2023).

**Checkpoint Selection and Early Stopping**. In supervised learning, checkpoint selection typically uses validation loss or accuracy for early stopping (Prechelt, 2002). For RL, aggregated rewards or held-out validation sets are common (Christiano et al., 2017; Stiennon et al., 2020; Liu et al., 2025a), but these can be misleading in non-stationary environments like LLM fine-tuning, where overfitting to training distributions is prevalent (Wei et al., 2022). Recent works explore online metrics, such as in-context learning performance (Brown et al., 2020), but require extra evaluations. Our approach differs by reusing per-sample rewards and uncertainty from the RL process itself, avoiding additional inference and focusing on hard samples for better generalization signals, inspired by data-centric views in machine learning (Liang et al., 2022; Jain et al., 2024).

**Sample Difficulty Estimation**. Estimating data sample difficulty is a long-standing problem in machine learning. Data cartography (Swayamdipta et al., 2020a) maps examples based on training

dynamics, using signals such as consistency (variance in correctness across epochs) or confidence. Uncertainty quantification in deep learning, often measured via entropy or log-likelihood (Gal & Ghahramani, 2016), has also been widely applied to identify hard or ambiguous samples in contexts such as active learning (Settles, 2009) and out-of-distribution detection (Ren et al., 2019). In large language models, uncertainty has proven useful for calibrating generations (Huang et al., 2025) and detecting hallucinations (Manakul et al., 2023). Unlike static, precomputed difficulty measures (Swayamdipta et al., 2020b), our approach treats difficulty as adaptive: evolving with the model's capabilities at each checkpoint. This dynamic perspective aligns with recent findings that performance on challenging samples is more predictive of generalization in reasoning tasks (Suzgun et al., 2023; Mirzadeh et al., 2024). Building on these insights, we use uncertainty-driven difficulty estimation to design a lightweight checkpoint scoring mechanism tailored to RL finetuning of LLMs.

## 3 METHOD

### 3.1 OVERVIEW

Our approach consists of 2 main stages: logging and checkpoint assessment, with details as follows.

**Logging** Our method assumes that, during RL fine-tuning, the training process logs per-sample information at every step. Specifically, at each training step, a batch of $B$ data samples (questions) is processed. For each sample $s$, the model generates $N$ answers, and the log-probabilities of each generated answer $a$, $\log p(a|s)$, are recorded along with their associated rewards $R$. This results in $N$ question–answer pairs per sample, each annotated with both reward and log-likelihood information, which forms the basis for our uncertainty-guided checkpoint scoring. We note that all the information is naturally obtained during each forward pass of the LLM, so no additional computation is required: only recording the logged rewards and log-probabilities. In standard RL training, the total or average reward across the current and prior batches is typically used to measure the quality of the current model. Rather than relying on batch-averaged rewards, we consider the reward for each sample, especially "harder" ones, providing a finer-grained and more discriminative signal for performance evaluation.

**Checkpoint Assessment** To assess a checkpoint $C$, performance is not assessed solely at that step; instead, a *training window* $[C.\text{step} - \delta, C.\text{step})$, which includes all steps from $C.\text{step} - \delta$ to $C.\text{step}-1$, is used to capture the most recent $\delta$ steps. The training window size $\delta$ is a hyperparameter chosen with respect to training dynamics and dataset size. Rather than requiring long histories, we show that even small windows (e.g., $\delta = 10$) capture most of the discriminative signal, making the method more practical for frequent monitoring. The effect of this hyperparameter will be investigated in Section 5.2. This window smooths out stochastic fluctuations of RL updates and provides a more stable signal for comparison across checkpoints. Within the window, the score is computed over the hardest $p\%$ of samples rather than across all question-answer pairs. This design focuses evaluation on the subset most indicative of generalization, while the $\delta$-step window ensures stability without added computational cost. We also provide the overview pseudo-code for our method in Algorithm 1. In the following sections, we present the details of the checkpoint assessment stage.

### 3.2 MEASURING THE DIFFICULTY OF A SAMPLE VIA MODEL UNCERTAINTY

Static difficulty labels (e.g., precomputed difficulty scores estimated by an external model (Luo et al., 2025; Do et al., 2025)) fail to reflect the model's evolving capabilities: a question may be "hard" for an early checkpoint but "easy" later. To avoid this pitfall, we propose dynamically quantifying the "hardness" of a data sample using uncertainty estimation.

Following prior works (Huang et al., 2025), we model LLM uncertainty on an input sample by approximating the predictive entropy as an average negative log-likelihood (ANLL) of the greedy-generated answer:

$$\text{ANLL}(a) = -\frac{1}{T} \sum_{t=1}^{T} \log p_\theta(a^t \mid a^{<t}, s), \tag{1}$$

---

**Algorithm 1** Uncertainty-Guided Checkpoint Selection (UGCS)

---

**Require:** Training log $\mathcal{L}$, checkpoint $C$. Hyperparameters: window size $\delta$, top-$p\%$, samples per question $N$

**Ensure:** Checkpoint score $\text{Score}(C)$
 1: Collect samples within the training window: $\mathcal{W}(C) \leftarrow \{s \in \mathcal{L} \mid s.\text{step} \in [C.\text{step} - \delta, C.\text{step})\}$
 2: **for** $s \in \mathcal{W}$ **do**
 3:    Initialize $R_s \leftarrow 0, \text{ANLL}_s \leftarrow 0$
 4:    **for** $i = 1$ to $N$ **do**
 5:       Collect answer $a_i$ with length $T$ from RL training
 6:       Compute $\text{ANLL}(a_i|s_i) = -\frac{1}{T} \sum_{t=1}^{T} \log p_\theta(a_i^t \mid a_i^{<t}, s_i)$
 7:       $R_s \leftarrow R_s + R_{s_i}, \text{ANLL}_s \leftarrow \text{ANLL}_s + \text{ANLL}(a_i|s_i)$
 8:    **end for**
 9:    Average: $R_s \leftarrow R_s/N, \text{ANLL}_s \leftarrow \text{ANLL}_s/N$
10: **end for**
11: Sort $\mathcal{W}(C)$ by $\text{ANLL}_s$ descending, select top-$p\% \rightarrow \mathcal{W}_p(C)$
12: $\text{Score}(C) = \frac{1}{|\mathcal{W}_p(C)|} \sum_{s \in \mathcal{W}_p(C)} R_s$
13: **return** $\text{Score}(C)$

---

where $s$ is the question, $a = (a^1, \ldots, a^T)$ is the greedy-generated answer of length $T$, $a^t$ is the $t$-th token in $a$, and $p_\theta$ is the model's predicted token distribution. This measure requires only a single forward pass, making it far more efficient while enabling per-sample uncertainty estimation at any point in training. Therefore, additional computation to calculate ANLL is unnecessary, as it is already obtained during RL training, whereas other methods, such as semantic entropy (Farquhar et al., 2024), would require extra computation. Beyond efficiency, uncertainty-based difficulty has two advantages: (1) it adapts dynamically to the model's current state, ensuring that "hardness" reflects present learning progress; and (2) it offers fine-grained resolution at the level of individual responses to the question, rather than relying on a fixed, per-question label. We compare ANLL against alternative difficulty metrics, such as consistency-based difficulty measures, in Section 5.2.

## 3.3 CHECKPOINT SELECTION CRITERION

To evaluate a checkpoint $C$, we compute a checkpoint score based on the hardest samples observed during its associated training window. Specifically, the interval $[C.\text{step} - \delta, C.\text{step})$, representing the recent $\delta$ training steps preceding checkpoint $C$, is considered. Within this window, we first identify the subset of samples that exhibit the highest difficulty for the model, as measured by their uncertainty values. Let $\mathcal{W}_p(C)$ denote the top-$p\%$ of these hardest samples in the window. That is, $\mathcal{W}_p(C)$ contains the most challenging inputs, ranked by uncertainty, reflecting areas where the model is least confident. The final score assigned to checkpoint $C$ is then computed based on this subset of top-$p\%$ hardest samples, effectively capturing the model's performance on the most difficult cases encountered in the recent training period. Formally, the score is defined as:

$$\text{Score}(C) = \frac{1}{|\mathcal{W}_p(C)|} \sum_{s \in \mathcal{W}_p(C)} R_s, \tag{2}$$

where $R_s$ is the average reward of sample $s$ over all $N$ answers within the set of $\mathcal{W}_p(C)$. This formulation ensures that checkpoint ranking is driven by performance on the most challenging data points, which are more likely to expose differences in generalization capability.

The parameter $p$ controls the trade-off between reliability and discriminative power: using a too small subset may produce unstable estimates, while including too many samples dilutes the focus on hard cases. For example, setting $p = 3$ means that only the 3% hardest of samples in the training window are considered when computing the checkpoint score. The value of $p$ is determined empirically and further explored in Section 5.2.

Because both $R_s$ and uncertainty are computed directly from the existing training logs, the entire scoring procedure requires no additional forward passes or dataset processing. This means our method introduces negligible overhead. Particularly, short $\delta$-step windows make it efficient enough

for adaptive checkpointing or early stopping in long training runs. In practice, we can either save several checkpoints regularly and later select the one with the highest $\text{Score}(C)$, or save only the checkpoint that currently has the highest score, designating it as the optimal model for deployment or further evaluation.

## 4 EXPERIMENTAL SETUP

### 4.1 TRAINING PROTOCOL

We fine-tune three tiny LLMs ($\leq$ 1B parameters), Qwen2.5-0.5B-Instruct (**Qwen2.5-0.5B**), Falcon3-1B-Instruct (**Falcon3-1B**), and **Qwen3-0.6B**, on a single NVIDIA H100 GPU. We choose these models for several reasons: (i) they are suitable for our hardware constraints, allowing efficient fine-tuning on a single GPU; (ii) despite their small size, they can exhibit high variance in performance, making them an informative testbed for evaluating our approach; and (iii) recent tiny LLMs, such as Qwen3-0.6B, are particularly strong as its reasoning capabilities are comparable to those of many 8B models (Yang et al., 2025), showing that even compact models can achieve competitive performance on complex reasoning tasks. Consequently, it is critical to identify the optimal checkpoint to fully leverage the maximum performance of these models after fine-tuning. To evaluate the robustness of checkpoint selection across different training scenarios, we conduct training experiments using three distinct datasets: **GSM8K** (Cobbe et al., 2021), **DeepScaleR** (Luo et al., 2025), and **GSM-symbolic** (Mirzadeh et al., 2024).

GSM8K contains 8.5K grade school math word problems (7.5K train, 1K test) designed for multi-step reasoning. DeepScaleR aggregates about 40K competition-level problems from AIME (Mathematical Association of America, 2024), AMC (Mathematical Association of America, 2023), Omni-MATH (Gao et al., 2024), and Still (Team et al., 2025), offering diverse coverage across domains and difficulty levels. GSM-symbolic is a template-based extension of GSM8K that introduces controlled variations to test robustness. For training, we use the full GSM8K (7.5K) and GSM-symbolic (5K) sets, while subsampling 2K examples from DeepScaleR to keep the data size practical and to reflect limited-data scenarios. Each training setup is run with three distinct random seeds to account for variability.

**Training details**. We finetune LLMs using the GRPO framework (Guo et al., 2025). The code is based on the Open-R1 repository (HuggingFace, 2025). We use batch size $B = 8$, maximum response length $L = 1200$, number of samples per question $N = 8$, and $E = 1000$ training steps. Checkpoints are saved every $\delta = 100$ steps (10 checkpoints in total). Full implementation details are provided in Appendix A.2.

### 4.2 EVALUATION PROTOCOL

We evaluate the models on four benchmarks that span a wide range of difficulty levels: MATH-500 (Lightman et al., 2023), Minerva Math (Lewkowycz et al., 2022), OlympiadBench (He et al., 2024), and AMC 2023 (Mathematical Association of America, 2023). MATH-500 is a curated subset of 500 competition-level problems covering topics like algebra, geometry, and number theory. Minerva Math (MINERVA) contains 272 undergraduate-level problems drawn from sources such as arXiv papers, emphasizing complex quantitative reasoning and symbolic manipulation. Olympiad-Bench (OLYMPIAD) comprises over 8K problems from international and national mathematics and physics Olympiads, marked by advanced difficulty and creative reasoning challenges. AMC 2023 (AMC23) includes 50 high school competition problems that test speed and accuracy under time constraints. These datasets range from high school competition problems to advanced Olympiad-level challenges, providing a comprehensive test of generalization.

We follow a zero-shot setting for all evaluations. Our evaluation framework is based on LightEval (Fourrier et al., 2023), employing its extractive match metric, which applies regex-based conditions to precisely extract and parse generated answers. Answers must adhere to a strict, predefined format to be successfully extracted; otherwise, they are counted as incorrect.

**Evaluation metric**. We run three different seeds for each training and report the accuracy of selected checkpoints on four benchmarks (mean±std) as the evaluation metric.

| Model | Method | MATH 500 | MINERVA | OLYMPIAD | AMC23 |
|---|---|---|---|---|---|
| Qwen2.5-0.5B | Train Reward | 19.7±0.1 | **1.7±0.3** | **3.3±0.5** | 2.5±2.0 |
| | Val Reward | **21.5±0.4** | **1.7±0.4** | 2.2±1.0 | 3.3±3.8 |
| | Last Checkpoint | 21.0±0.9 | 1.6±0.7 | 2.7±1.3 | 6.7±2.9 |
| | $p\%$ Reward | 20.7±1.0 | 1.6±0.5 | 1.3±0.5 | 5.8±4.7 |
| | UGCS (Ours) | 20.3±0.4 | 1.5±0.0 | **3.3±1.6** | **7.5±6.1** |
| Qwen3-0.6B | Train Reward | **75.4±0.0** | 13.7±0.5 | 23.8±1.7 | 47.5±2.0 |
| | Val Reward | **75.4±0.0** | 12.9±1.0 | 24.2±2.7 | **52.5±13.9** |
| | Last Checkpoint | 75.1±0.7 | 13.6±2.2 | **25.6±1.6** | 47.5±4.3 |
| | $p\%$ Reward | **75.4±0.0** | 13.4±0.3 | 24.9±0.6 | 49.2±6.2 |
| | UGCS (Ours) | **75.4±0.0** | **14.7±0.9** | 25.1±2.7 | 50.8±1.2 |
| Falcon3-1B | Train Reward | 18.5±0.9 | 2.9±1.1 | **2.4±0.6** | 5.0±4.1 |
| | Val Reward | 18.2±1.4 | 3.6±0.4 | 2.2±0.8 | 3.3±1.4 |
| | Last Checkpoint | 17.5±0.9 | **4.8±0.9** | 1.8±1.0 | 4.1±1.4 |
| | $p\%$ Reward | 18.4± 0.9 | 2.9±0.5 | 2.0±0.5 | **6.7±1.2** |
| | UGCS (Ours) | **20.2±0.6** | 3.8±0.2 | **2.4±1.1** | 5.8±1.2 |

Table 1: Performance comparison across various LLMs and datasets using **GSM8K** as training data. Bold indicates the best score, with ties (Cohen effect size <0.5) also bolded.

## 4.3 BASELINES

All baselines, except the one that utilizes the validation set, leverage samples within the same training window of the last $\delta$ training steps to ensure fairness. We set the training window $\delta = 100$ for the main results and investigate the effect of this hyperparameter in Section 5.2. The baselines include: (i) **Train Reward**: average reward over training samples within the window; (ii) **Val Reward**: average reward over a validation set containing samples separate from the training set. We have three validation sets corresponding to GSM8K, DeepScaleR, and GSM-symbolic. This baseline is often the standard way to select optimal checkpoints in machine learning, yet in LLM finetuning, it requires costly inference for every checkpoint evaluation, which significantly increases the overall training time. (iii) **Last Checkpoint**: evaluation results reported at the last training checkpoint (i.e., step 1000th in our setting). This is commonly used in practice to select the deployment checkpoint. (iv) **p% Reward**: average reward over top-$p\%$ samples with highest rewards. The value of $p$ is specified for each model and matches the $p$ used in our method. This method is similar to ours, except that it relies solely on reward (accuracy) to measure the top hardest samples, without incorporating uncertainty. (v) **UGCS (Ours)**: average reward over only the top $p\%$ hardest samples, ordered by mean ANLL across $N = 8$ generations during training. The uncertainty metrics are computed on-the-fly from the model during the training process. Choices of $p$ will be discussed in Section 5.2.

For each baseline, we calculate the corresponding checkpoint scores for all checkpoints based on saved training logs or validation results. To ensure a fair comparison, we use validation sets of the same size as the training window $\delta$. For each validation sample, we generate 8 completions and use their average reward to compute the final checkpoint metric. The evaluation accuracy for the checkpoints with the highest checkpoint scores is reported as the final results for each baseline.

## 5 EXPERIMENTAL RESULTS

## 5.1 REASONING BENCHMARK

Tables 1, 2, and 3 present performance across all training datasets: GSM8K, DeepScaleR and GSM-symbolic, respectively. Training with GSM8K (Table 1), our method achieves the best performance in 6 out of 12 cases, outperforming the second-best approaches (Train Reward and Val Reward, 4/12) by roughly 0.5–1.5% on average across all benchmarks. In contrast, the $p\%$ Reward method identifies the top checkpoint in only 2 out of 12 cases, highlighting the added value of our uncertainty-based criteria. Similarly, the Last Checkpoint performs poorly, achieving the best result in just 2 out of 12 cases.

| Model | Method | MATH 500 | MINERVA | OLYMPIAD | AMC23 |
|---|---|---|---|---|---|
| Qwen2.5-0.5B | Train Reward | 22.0±0.4 | 1.1±0.5 | 2.0±0.8 | 7.5±2.5 |
| | Val Reward | 21.4±1.7 | 1.1±0.3 | 1.3±0.3 | 3.3±1.2 |
| | Last Checkpoint | 20.5±1.1 | **2.1±0.8** | 1.6±1.0 | 7.5±5.0 |
| | $p\%$ Reward | 22.0±0.8 | 0.7±0.3 | 2.0±0.3 | **10.0±0.0** |
| | UGCS (Ours) | **22.6±1.1** | 1.1±0.3 | **3.3±1.1** | **10.0±0.0** |
| Qwen3-0.6B | Train Reward | 76.4±1.5 | 12.5±0.9 | 26.0±1.8 | 50.0±2.5 |
| | Val Reward | 76.4±2.4 | 14.8±1.2 | 25.1±2.1 | 47.5±6.6 |
| | Last Checkpoint | 75.1±0.9 | 14.5±2.8 | 26.9±2.3 | 54.2±5.2 |
| | $p\%$ Reward | 76.0±1.2 | **15.2±0.8** | 24.7±1.5 | 45.0±2.5 |
| | UGCS (Ours) | **78.4±1.8** | 14.3±1.2 | **28.7±0.7** | **55.0±2.1** |
| Falcon3-1B | Train Reward | **19.6±0.9** | 2.2±0.3 | 1.3±0.3 | 5.0±1.7 |
| | Val Reward | 17.5±1.1 | 3.8±0.4 | 1.1±0.4 | 5.0±2.5 |
| | Last Checkpoint | 17.5±1.2 | 3.4±0.2 | **2.9±1.4** | 5.0±1.2 |
| | $p\%$ Reward | 18.8±1.1 | 3.7±0.3 | 2.0±0.7 | 7.5±1.2 |
| | UGCS (Ours) | **19.6±0.5** | **4.0±0.5** | 2.0±1.1 | **10.0±2.0** |

Table 2: Performance comparison across various LLMs and datasets using **DeepscaleR** as training data. Bold indicates the best score, with ties (Cohen effect size <0.5) also bolded.

| Model | Method | MATH 500 | MINERVA | OLYMPIAD | AMC23 |
|---|---|---|---|---|---|
| Qwen2.5-0.5B | Train Reward | 20.5±0.7 | 1.6±0.3 | 2.0±0.9 | 5.8±2.4 |
| | Val Reward | 19.7±0.6 | 2.2±0.4 | 2.6±1.4 | 5.0±2.5 |
| | Last Checkpoint | 19.9±0.7 | **2.5±0.7** | 3.1±0.4 | **6.7±5.2** |
| | $p\%$ Reward | 19.7±0.5 | 2.2±0.3 | **3.8±1.1** | 5.0±2.0 |
| | UGCS (Ours) | **21.4±0.7** | **2.5±0.6** | 3.1±0.8 | 5.8±2.4 |
| Qwen3-0.6B | Train Reward | 74.9±1.9 | 13.0±0.6 | **26.7±2.5** | 50.0±7.1 |
| | Val Reward | 75.4±0.2 | 13.2±1.7 | 25.1±1.4 | 52.5±2.5 |
| | Last Checkpoint | 75.8±1.3 | **14.7±1.0** | 25.6±3.1 | 47.5±4.3 |
| | $p\%$ Reward | 74.5±1.2 | **14.7±1.0** | 26.2±2.3 | 48.3±6.6 |
| | UGCS (Ours) | **76.1±1.0** | 14.3±0.5 | 24.4±2.5 | **56.7±4.7** |
| Falcon3-1B | Train Reward | 17.7±0.6 | 3.8±0.5 | 2.2±1.1 | 3.3±1.2 |
| | Val Reward | 18.2±0.9 | 3.1±0.6 | **2.7±1.8** | 5.8±1.4 |
| | Last Checkpoint | 16.3±4.0 | 3.1±1.3 | 1.1±0.4 | 4.2±1.4 |
| | $p\%$ Reward | 17.8±0.4 | **4.0±0.8** | 1.0±0.3 | 3.3±3.1 |
| | UGCS (Ours) | **18.3±1.7** | 3.8±0.3 | 2.3±1.7 | **6.7±5.1** |

Table 3: Performance comparison across various LLMs and datasets using **GSM-symbolic** as training data. Bold indicates the best score, with ties (Cohen effect size <0.5) also bolded.

When using DeepScaleR as training data (Table 2), our method consistently outperforms the second-best baselines with average improvements of 1.1-2.9%, successfully finding the top checkpoints in 9/12 cases. Notably, on AMC23, gains range from 2.5-5% over all baselines, highlighting its effectiveness on competition-level problems. Interestingly, the Last Checkpoint baseline ranks as the second-best method here, achieving the top performance in only 2 out of 12 cases, still far behind our approach. The supposedly strong Val Reward baseline performs poorly, achieving 0 out of 12 wins, highlighting the complexity of checkpoint selection in RL finetuning of LLMs.

When using GSM-symbolic (Table 3), our method remains the top performer, achieving the highest accuracy in 5 out of 12 cases, while the second-best baseline, Last Checkpoint, wins in 3 out of 12 cases. At the model level, our approach performs on par with other checkpoint selection methods for Falcon3-1B, surpasses them on Qwen2.5-0.5B with an average gain of 3% across the four datasets, and delivers particularly strong improvements on Qwen3-0.6B, reaching up to 6.7% gain on AMC23. Thus, our method is particularly beneficial for high-capability LLMs, enabling them to fully leverage their strengths and achieve state-of-the-art results.

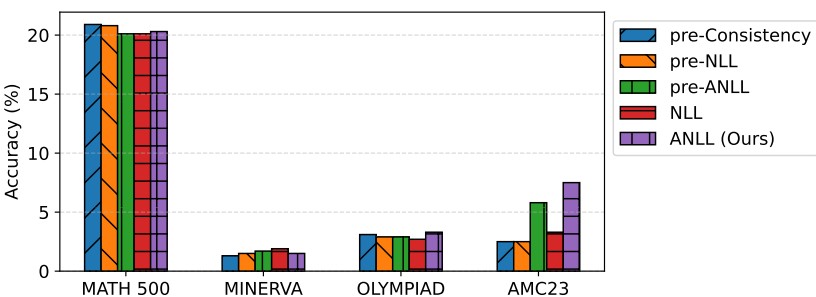

Figure 1: Mean Accuracy across datasets for different difficulty metrics on Qwen2.5-0.5B and GSM8K training.

## 5.2 ABLATION STUDIES

In this section, we conduct a series of ablation studies to analyse the effect of each component and examine how alternative configurations influence final performance. We focus on three main aspects: (1) difficulty metrics, (2) choice of $p$, and (3) effect of training window size $\delta$. To reduce computational overhead, we only experiment over Qwen2.5-0.5B using GSM8K training dataset.

### 5.2.1 DIFFICULTY METRICS

Our method relies on estimating question difficulty to prioritize challenging samples for checkpoint evaluation. We compare five difficulty metrics: (i) our proposed metric: average negative log-likelihood (*ANLL*) where we compute the ANLL on-the-fly during training to estimate the difficulty of the sample with respect to the LLM at the particular training step; (ii) negative log-likelihood (*NLL*), a common uncertainty metric computed during training, also computed on-the-fly as above; (iii) a consistency-based difficulty measure (*pre-Consistency*) (Swayamdipta et al., 2020b), which estimates difficulty via correctness variance across multiple generations of a fixed, external LLM; (iv) precomputed average negative log-likelihood (*pre-ANLL*); and (v) precomputed negative log-likelihood (*pre-NLL*), where difficulty scores are calculated once before training and remain fixed. This comparison isolates the roles of adaptivity and efficiency in difficulty estimation. All metrics are computed using the Qwen2.5-0.5B model, with difficulty scores derived following the methodology in Shi et al. (2025).

Figure 1 shows the impact of different difficulty metrics on evaluation performance when fine-tuning on the GSM8K dataset. Our proposed metric (ANLL) consistently performs competitively, achieving the highest scores on OlympiadBench and AMC 2023, and near-top performance on MATH-500. This suggests that ANLL effectively identifies challenging samples that correlate with generalization. Comparing on-the-fly uncertainty measures, the averaging in ANLL, which normalizes uncertainty across sequence lengths, appears to provide a more robust signal for ranking question difficulty, particularly for datasets with varied response lengths like AMC 2023. Additionally, these measures outperform precomputed metrics, highlighting the importance of adaptive difficulty estimation that evolves with the model's training dynamics. Since our metric is computed directly from training logs without extra LLM forward passes, it offers a practical and efficient solution for checkpoint selection in real-world RL settings.

### 5.2.2 CHOICE OF $p$

By design, our checkpoint evaluation focuses on the $p\%$ hardest samples within each training window, trading off *reliability* (larger $p$, smoother estimates) against *discriminative power* (smaller $p$, sharper focus on hard samples). To quantify this trade-off, we vary $p$ from 1 to 20 and score fine-tuned checkpoints on GSM8K using MATH-500 as the calibration set. MATH-500 is chosen because it is the least challenging among the four evaluation datasets (MATH-500, Minerva Math, OlympiadBench, and AMC 2023), ensuring that all checkpoints achieve reasonable performance. For each model–training dataset pair, we select the value of $p$ that yields the highest accuracy on MATH-500.

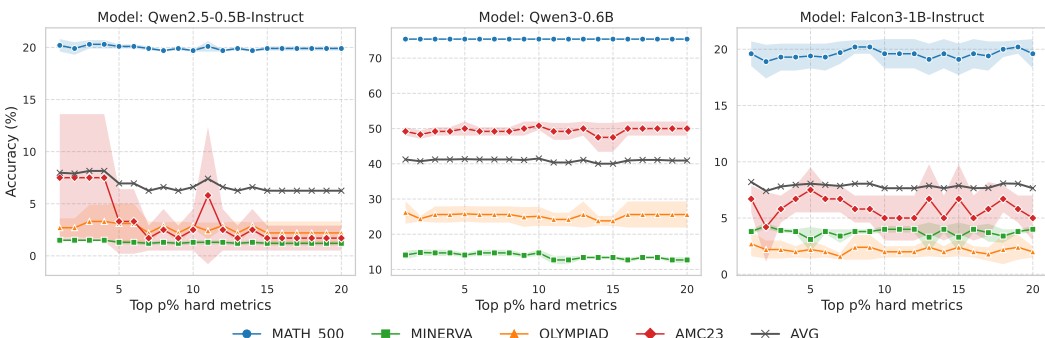

Figure 2: Results on GSM8K training when adjusting $p$ for various models and datasets. For each model, AVG denotes the average accuracy over four datasets, while the others (with shaded ranges) represent individual datasets with their standard deviations.

Figure 2 shows the effect of $p$ when fine-tuning on GSM8K. For Qwen2.5-0.5B, smaller $p$ (e.g., $p = 3$) maximizes discriminative power by emphasizing the hardest samples, but yields larger variation across evaluation datasets, reflecting lower reliability. For stronger models such as Qwen3-0.6B, the performance curves are nearly flat across $p$, so reliability is less of a concern. In this case, a moderate value such as $p = 10$ provides the best balance: it avoids the instability of very small subsets (e.g., $p = 1$), while still focusing on challenging samples and preventing dilution of the metric by easier cases. Overall, weaker LLMs benefit from smaller $p$, which sharpens discrimination even if reliability drops, while stronger models favor moderate $p$, which preserves stability without diluting the metric.

Based on this trend, we select $p = 3$ for Qwen2.5-0.5B, $p = 10$ for Qwen3-0.6B, and $p = 8$ for Falcon3-1B when training on GSM8K. Without calibration, we generally recommend $p = 3$ for weaker LLMs and $p = 10$ for stronger ones, which remain close to calibrated performance. The complete set of selected values used in Section 5.1 is reported in Appendix A.1.

### 5.2.3 EFFECT OF TRAINING WINDOW SIZE $\delta$

We examine how the training window size used for aggregating rewards affects evaluation performance. Figure 4 in the Appendix shows the benchmarking results across four datasets when fine-tuning Qwen2.5-0.5B on GSM8K. Our ablations indicate that small training windows (e.g., $\delta = 10$ steps) already capture most of the useful signal, yielding comparable performance to larger windows such as $\delta = 100$. On MATH-500, performance remains stable across different $\delta$ values, and fluctuations on other datasets are minor, except for AMC 2023. This finding highlights that the full training history is unnecessary; shorter windows enable more frequent checkpoint evaluation and support adaptive stopping during long RL runs.

## 6 CONCLUSION

We presented Uncertainty-Guide Checkpoint Selection (UGCS), a validation-free checkpoint selection method that ranks RL fine-tuned models based on performance on the hardest, most uncertain samples. By focusing on challenging cases within a short training window, UGCS provides a stable and discriminative signal without extra forward passes or computational cost. Experiments across multiple reasoning datasets show that UGCS consistently identifies checkpoints with superior generalization, particularly on difficult tasks. This demonstrates that a model's ability to reliably solve its hardest problems is a strong indicator of overall robustness. UGCS offers a simple, efficient, and broadly applicable alternative to traditional reward- or validation-based checkpointing.

## REPRODUCIBILITY STATEMENT

We have submitted the source code separately for the reviewing process. Upon publication, we will release the implementation as open-source with the necessary instructions to ensure reproducibility.

## LLM USAGE

Large Language Models (LLMs) were not involved in the design, implementation, or analysis of our method. They were only used to refine the presentation of the paper by correcting grammar and improving writing clarity.

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

# A APPENDIX

## A.1 VALUES OF $p$

Table 4 summarizes the selected values of $p$ for all model–dataset combinations based on validation performance on MATH-500; these values are used in the main results reported in Section 5.1.

| Training dataset | Model | Value of $p$ |
|---|---|---|
| GSM8K | Qwen2.5-0.5B | 3 |
| | Qwen3-0.6B | 10 |
| | Falcon3-1B | 8 |
| DeepScaleR | Qwen2.5-0.5B | 5 |
| | Qwen3-0.6B | 5 |
| | Falcon3-1B | 2 |
| GSM-symbolic | Qwen2.5-0.5B | 16 |
| | Qwen3-0.6B | 7 |
| | Falcon3-1B | 8 |

Table 4: Selected values of $p$ for each model–training dataset combination, chosen based on validation accuracy on MATH-500.

## A.2 IMPLEMENTATION DETAILS

### A.2.1 SYSTEM PROMPT

Following (HuggingFace, 2025), the system prompt asks the model to generate the answer with clear requirements, with reasoning and answer following the format, as described in Figure 3.

---
SYSTEM PROMPT
You are a helpful assistant. Please reason step by step, and put your final answer within \boxed{}.

---

Figure 3: System prompt used in our experiments.

### A.2.2 TRAINING HYPERPARAMETERS

In this section, we provide the training details of UGCS in Table 5.

| Parameters | Value |
|---|---|
| Batch size ($B$) | 8 |
| Number of samples per question ($N$) | 8 |
| Maximum completion length ($L$) | 1200 |
| Number of training epochs ($E$) | 1000 |
| Initial learning rate ($\alpha$) | $5e^{-6}$ |
| Weight Decay | 0.1 |
| Warmup Ratio | 0.1 |
| lr_scheduler_type | cosine |
| Adam $\beta_1$ | 0.9 |
| Adam $\beta_2$ | 0.99 |
| bf16 | True |
| Gradient accumulation steps | 8 |
| Max grad norm ($G_{norm}$) | 0.1 |
| $\epsilon$ | $1e^{-6}$ |

Table 5: Parameters used in fine-tuning process.

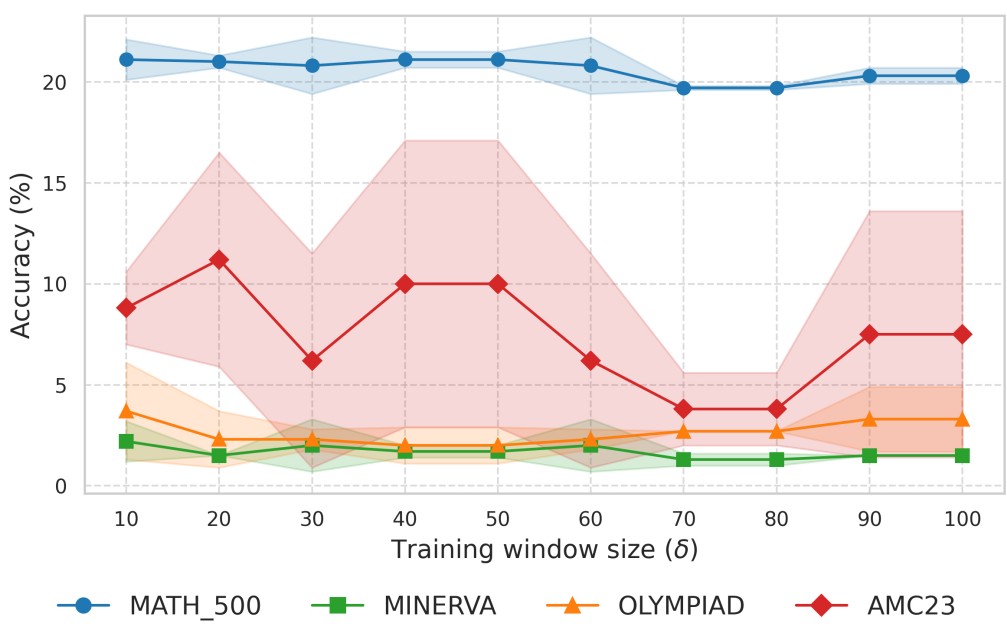

Figure 4: Results on GSM8K training across multiple datasets when varying $\delta$.

### A.2.3 MODEL AND DATA REFERENCES

We list the links to the LLM models and datasets in Table 6.

| Models/Datasets | URL |
|---|---|
| Qwen2.5-0.5B-Instruct | `https://huggingface.co/Qwen/Qwen2.5-0.5B-Instruct` |
| Qwen3-0.6B | `https://huggingface.co/Qwen/Qwen3-0.6B` |
| Falcon3-1B-Instruct | `https://huggingface.co/tiiuae/Falcon3-1B-Instruct` |
| GSM8K | `https://huggingface.co/datasets/openai/gsm8k` |
| GSM-symbolic | `https://github.com/apple/ml-gsm-symbolic` |
| DeepScaleR | `https://huggingface.co/datasets/lime-nlp/DeepScaleR_Difficulty` |
| MATH-500 | `https://huggingface.co/datasets/HuggingFaceH4/MATH-500` |
| MINERVA | `https://huggingface.co/datasets/math-ai/minervamath` |
| OLYMPIAD | `https://huggingface.co/datasets/Hothan/OlympiadBench` |
| AMC23 | `https://huggingface.co/datasets/knoveleng/AMC-23` |

Table 6: Models and Datasets Details.

