# OpenReview forum: "Uncertainty-Guided Checkpoint Selection  for Reinforcement Finetuning of Large Language Models"
_ICLR.cc/2026/Conference — ICLR 2026 Conference Withdrawn Submission_

### Official Review · Reviewer_7zXd · 2025-10-25

**Soundness:** 2
**Presentation:** 3
**Contribution:** 2
**Rating:** 2
**Confidence:** 4

**Summary:**

The proposed method, Uncertainty-Guided Checkpoint Selection (UGCS), identifies the best checkpoint during RL finetuning of LLMs without using validation data. It computes a checkpoint score based on how well the model performs on its most uncertain training samples.

At each step, the model’s uncertainty for each generated answer is measured using average negative log-likelihood (ANLL), which is already available from training logs. Within a short recent window of training steps, UGCS selects the top-p% most uncertain samples (the hardest cases) and averages their rewards. This average reward becomes the checkpoint’s quality score. This method does not add any additional cost.

**Strengths:**

1. UGCS cleverly reuses uncertainty and reward information already available in training logs, avoiding any additional forward passes or validation data—making it both efficient and practical.

2. By emphasizing performance on the most uncertain samples, it highlights checkpoints that have truly improved at hard cases rather than just memorizing easy ones.

3. The method’s design smooths out RL noise through short training windows, and the paper itself explains it clearly with clean, well-organized writing that makes the idea easy to follow.

**Weaknesses:**

1. In Algorithm 1, clearly mention that (R_{si}) is the reward of the i-th generated answer for sample s. It is used but not defined.

2. The evaluation rule says answers must strictly follow a predefined format; otherwise, they are marked incorrect. In practice, do models actually follow this format? Please report what percentage of outputs are rejected only because of formatting errors.

3. It would be helpful to add another column to Tables 1, 2, and 3 showing the average accuracy across all datasets for each checkpoint selection method and model.

4. Figure 1 is difficult to interpret because the bars are very close in height. Adding the exact accuracy values above each bar would make it much clearer which method performs best.

5. In Figure 2, I don’t see any meaningful difference across different p% values. I would have expected performance to first increase and then decrease. The flat curves don’t really show that focusing on hard (uncertain) examples helps.

6. I would like to also see a random-selection baseline. Please show what happens if you randomly choose p% of training samples for evaluation instead of using uncertainty. This would make it clearer whether the observed trend actually comes from selecting uncertain samples.

7. I also cannot infer the claim “weaker LLMs benefit from smaller p while stronger models favor moderate p” from Figure 2. The figure doesn’t seem to support that statement.

8. In the paragraph that says “Based on this trend, we select p = 3 for Qwen2.5-0.5B…” it sounds like p is chosen based on test-set performance (MATH-500). Please clarify this, because it may look like using test data for tuning.

**Questions:**

Please take a look at the weaknesses section. I would also appreciate it if you could help me interpret the results. I might be misunderstanding them, but many of the reported results don’t seem very convincing and don’t clearly support the paper’s main claims.

---

### Official Review · Reviewer_iHrm · 2025-10-29

**Soundness:** 2
**Presentation:** 2
**Contribution:** 1
**Rating:** 2
**Confidence:** 3

**Summary:**

This paper introduces a method called Uncertainty-Guided Checkpoint Selection (UGCS) for selecting optimal checkpoints during the reinforcement learning of LLMs. The core innovation of this approach lies in leveraging per-sample uncertainty and reward signals that are already available during training to dynamically identify the most difficult samples. UGCS computes the average reward over these hardest samples and uses it as a checkpoint quality score.

**Strengths:**

1. By assessing the model's performance on the most uncertain samples, it serves as a proxy for its generalization ability, providing a lightweight perspective on evaluating the RL fine-tuning process with a simple design.
2. Although the evaluation scope is limited, the mathematical reasoning tasks tested provide preliminary empirical evidence for the effectiveness of this method.

**Weaknesses:**

1. Potential Contradiction with the "Validation-Free" Claim: The method introduces key hyperparameters (p and δ) that rely on a "calibration set" to determine their optimal values. This set is functionally similar to a validation set, which undermines the core "validation-free" advantage of the approach.
2. Limited Evidence of Generalization: The experiments are confined to a single task domain (mathematical reasoning) and smaller-scale models. It remains unclear whether the findings can be generalized to other tasks (such as dialogue or creative writing) and larger-scale models.
3. Sensitivity to Preconditions: The method's effectiveness is contingent on certain assumptions. If the reward signals during training are noisy or the uncertainty estimations are unstable (particularly in the early training phases), the UGCS scoring mechanism could yield misleading results. Likewise, it may underperform on datasets that are either of very high quality or highly homogeneous.
4. Lack of Theoretical Grounding: While the paper offers a heuristic and intuitive motivation, it lacks a rigorous theoretical analysis to explain why and under what conditions performance on difficult samples serves as a reliable predictor of generalization. A formal theoretical framework would significantly strengthen the method's convincingness.

**Questions:**

1. Regarding Alternative Uncertainty Metrics: Have you considered or compared UGCS with other uncertainty quantification methods, such as Bayesian approximations (e.g., MC Dropout) or semantic entropy?
2. From Post Hoc Selection to Online Guidance: UGCS currently operates as a post hoc tool for checkpoint selection. Have you considered using its signal online to dynamically guide the training process itself? For instance, could the performance on difficult samples be used to adaptively adjust the learning rate, alter batch composition (i.e., for curriculum learning), or even trigger early stopping?
3. Scalability with Model Size: How do you expect the effectiveness of UGCS and the optimal choice of its hyperparameters (p and δ) to change as the model size increases significantly (e.g., scaling from 1B to 70B parameters or larger)?
4. Generalization to Other Tasks: Can the method be directly transferred to tasks beyond mathematical reasoning, such as dialogue safety alignment or code generation? What challenges, if any, do you foresee in applying UGCS to these domains?
5. Computational Overhead: Did you measure the practical computational overhead introduced by UGCS compared to the other baseline methods? If so, could you provide details on the cost?

---

### Official Review · Reviewer_DqzJ · 2025-10-31

**Soundness:** 3
**Presentation:** 3
**Contribution:** 2
**Rating:** 4
**Confidence:** 3

**Summary:**

The paper proposes UGCS (Uncertainty-Guided Checkpoint Selection) for RL fine-tuning of LLMs. Instead of using training/validation averages or picking the last checkpoint, UGCS ranks checkpoints by the mean reward on the top-p% most uncertain (hardest) samples within a short $\delta$-step window, using average negative log-likelihood (ANLL) as the per-sample uncertainty proxy. The method claims no extra forward passes, reusing logged token-level log-probs and rewards from training, and reports consistent improvements across tiny models.

**Strengths:**

1. Simple, low-overhead signal: Uses already-logged per-token log-probs and rewards; no extra inference loops. The $\delta$-window idea is practical for stabilizing noisy RL updates.
2. Focus on “hard” examples: The insight that performance on hard samples is more predictive of generalization is sensible and supported by prior “hard-sample”/data-cartography literature.
3. Clear algorithm and setup: UGCS is specified with pseudo-code (Alg. 1); training/eval details and the LightEval protocol are documented; hyperparameters are listed.
4. Ablation studies on uncertainty metric, p, and $\delta$ lend credibility

**Weaknesses:**

(Please respond to the questions directly)

1. concerns on the novelty
2. Limited scale of the experiment
3. Some details on the experiment design are of concern

**Questions:**

1. The main contribution of the methodology is applying ANLL-based hard-sample filtering for checkpoint selection during RLHF/GRPO; interesting but I'm not very clear if this is the correct approach. What happens if you rank “hardest” by lowest reward instead of by uncertainty? This is a crucial ablation to separate the value of uncertainty from the “focus on hard cases” idea.
2. Is ANLL computed on greedy outputs or on the N sampled trajectories generated during RL? If greedy, where do those come from without extra forward passes? If sampled, why is section 3.2 phrased as greedy? I'm a bit confused...
3. All experiments are on ≤1B models with math datasets. It’s unclear if the gains persist for larger models/tasks (code, safety, dialog). Any results on larger models or non-math tasks (safety, dialog)?
4. For the reuslts in Table 1, many numbers are small (1–3%) and likely within noise. “Bold indicates best; ties if Cohen’s d < 0.5” is an odd rule to me; Could you try reporting checkpoint-selection accuracy (how often UGCS picks the true best checkpoint ex-post) to quantify selection quality?
5. If $\delta$=10 already works, why fix checkpoint saves at 100 steps? Would more frequent saving further help selection stability? Another thing is that you pick p by maximizing accuracy on MATH-500; how do results change if p is tuned on a separate calibration set or via cross-validation?

---

### Official Review · Reviewer_i2ri · 2025-10-31

**Soundness:** 3
**Presentation:** 3
**Contribution:** 2
**Rating:** 2
**Confidence:** 3

**Summary:**

A checkpoint is a snapshot of a model’s parameters at an intermediate training stage.  Properly selected checkpoints can be leveraged to select a model that offers the best performance vs generalization trade-off. The paper proposes a new method for checkpoint selection that is based on a measure of model uncertainty. In the proposed method the performance of a checkpoint choice starts by identifying  the "hardest" question–answer pairs (based upon average negative log-likelihood) in a time-window. Then, a score for the checkpoint choice is computed as the average reward for a percentile of the hardest question–answer pairs. Results from a computational testbed provide evidence for the benefits of the proposed checkpoint selection mechanism.

**Strengths:**

Large-scale LLM training is noisy, nonconvex, and extremely expensive. Training longer does not guarantee a better model.
Thus, checkpoint selection helps to prevent overtraining (in supervised fine-tuning) and ensure alignment robustness (in RLHF/DPO).
The paper proposes a checkpoint selection methods that combines performance (reward) with a measure of uncertainty (ANLL).

**Weaknesses:**

Due to serial correlation, the sampled responses from the model within the training window are themselves correlated.
Thus, appropriate checkpoint selection must account for correlation in the computation of the score.
Otherwise, by making performance trends look more significant than they are, a form of evaluation bias is likely introduced.

**Questions:**

1. Please discuss the implications of serial correlation when you collect samples within the training window.

---

### Note · Authors · 2025-11-13

I have read and agree with the venue's withdrawal policy on behalf of myself and my co-authors.